# Trend Prediction Based on Multi-Modal Affective Analysis from Social Networking Posts

Kazuyuki Matsumoto *, Reishi Amitani, Minoru Yoshida and Kenji Kita

Graduate School of Sciences and Technology for Innovation, Tokushima University, Tokushima 770-8506, Japan
* Correspondence: matumoto@is.tokushima-u.ac.jp

**Abstract:** This paper propose a method to predict the stage of buzz-trend generation by analyzing the emotional information posted on social networking services for multimodal information, such as posted text and attached images, based on the content of the posts. The proposed method can analyze the diffusion scale from various angles, using only the information at the time of posting, when predicting in advance and the information of time error, when used for posterior analysis. Specifically, tweets and reply tweets were converted into vectors using the BERT general-purpose language model that was trained in advance, and the attached images were converted into feature vectors using a trained neural network model for image recognition. In addition, to analyze the emotional information of the posted content, we used a proprietary emotional analysis model to estimate emotions from tweets, reply tweets, and image features, which were then added to the input as emotional features. The results of the evaluation experiments showed that the proposed method, which added linguistic features (BERT vectors) and image features to tweets, achieved higher performance than the method using only a single feature. Although we could not observe the effectiveness of the emotional features, the more emotions a tweet and its reply match had, the more empathy action occurred and the larger the like and RT values tended to be, which could ultimately increase the likelihood of a tweet going viral.

**Keywords:** multi-modal buzz prediction; information diffusion; affective analysis

## 1. Introduction

In recent years, there has been a daily phenomenon in which topics and news about certain things spread rapidly on the Internet, triggered by comments posted on social networks. This phenomenon is called "buzz," but it is difficult to predict in advance what type of statements will cause a "buzz phenomenon", because various factors are involved. By detecting the signs of a buzz phenomenon as early as possible, companies can keep up with trends and utilize the buzz forecast results in their marketing activities.

Studies on trend analysis and buzz forecasting on social networks have been conducted for some time.

Lin et al. [1] predicted dynamic trends in social networks. They defined relevant concepts to quantify the dynamic characteristics of trends in social networks and proposed a dynamic activity-based trend prediction algorithm based on activity levels. The proposed method was validated on a DBLP network and achieved higher performance than the conventional method. The method used by Lin et al. differs from ours in that it does not focus on the content of posts, but, rather, on trend prediction by dynamically analyzing the relationships among users in the social network.

Noting the viral nature of posts that cause intense interaction in a short period, Deusser et al. [2] stated that early detection of the buzz phenomenon could help detect and address the negative effects of negative opinions directed at companies and individuals on social networking sites. They investigated the properties of the buzz phenomenon using logistic regression on a labeled set of over 100,000 posts on Facebook. They also constructed a

classifier based on a machine learning algorithm that can discriminate whether a post is a buzz. This classifier achieved high reproducibility and moderate accuracy. They also analyzed the classification results in detail and stated that the main factors of the buzz phenomenon are the number of comments from passive users, the number of "likes" on the comments, and the length of time the topic has been discussed. Our study differs in that it does not use user information to predict the buzz phenomenon; however, it is similar in that the buzz phenomenon is predicted by a machine learning algorithm.

Jansen et al. [3] constructed a buzz-detection system (BDS). They trained a model for use in the BDS using approximately 120,000 posted texts on Facebook as the training data. As a result, they found that the number of times a passive user engages with a Buzz post and the number of "likes" given to a comment are important as features for prediction. They achieved a buzz detection rate of >97% in the evaluation experiments. Their proposed BDS requires information, such as the number of "likes" and previous actions of users involved in the buzz for buzz prediction. This is not pure information at the time of posting, but a large element of information obtained after posting; therefore, the system does not work as a system to check the content before posting. The objective of this study was to predict the number of likes and retweets (RTs) using the content of the tweet at the time of posting and the attached image as the main features.

Amitani et al. [4] constructed a model to discriminate buzz tweets from text with images posted on Twitter using multi-task learning. Their method is similar to the present study in that it uses only the information at the time of tweet posting. However, their proposed method is superior to our method in that it can predict the scale of each numerical value, such as the number of likes and RTs, because their model classifies tweets into simple binary values (buzz or non-buzz). However, it cannot predict the scale of numerical values.

In this study, we propose a model that predicts the number of "likes" and RTs by detecting the most popular tweets on Twitter and analyzing the content of the body text, attached images, and replies to these tweets. This model predicts what types of tweets are likely to be spread by analyzing the content of the tweets themselves, eliminating personal factors, such as what types of tweets the poster has posted in the past, the attributes of the poster, and his/her name recognition.

## 2. Materials and Methods

### 2.1. Feature Vector Extraction Based on Pre-Trained Model

In this subsection, we discuss feature extraction from pre-trained models. Feature extraction from the text of tweets and replies to tweets, feature extraction from attached images, and extraction of affective features are also described.

#### 2.1.1. Corpus of Japanese Spoken Language BERT

Recent research in natural language processing has used bidirectional encoder representations from transformers (BERT), XLNet [5], BART [6], ALBERT [7], ELECTRA [8], RoBERTa [9], or T5 [10], and other large and complex neural network-based language models. These language models, known as general-purpose language models, are trained on general-purpose tasks using large training datasets to obtain general-purpose knowledge that can be applied to a variety of tasks. It has been shown that fine-tuning a small number of task-specific datasets, based on these general-purpose language models, saves time and costs, compared with training from scratch, and achieves higher performance.

A pre-trained BERT model of spoken Japanese (CSJ-BERT model) [11] was used to extract features from the text of posted and reply tweets. This model was developed and published by Retriva Inc.(Tokyo, Japan) Based on a corpus of spoken Japanese, two tasks, the masked language model and next sentence prediction, were performed using BERT.

The model trains only the grammatical part of the written BERT using a corpus of spoken Japanese and, then, performs domain adaptation on the written BERT using spoken data. By using the part of the layer that can acquire grammar and field adaptation, it is possible to acquire features with different characteristics from those of the middle layer.

Compared to written BERT and models with additional training on all layers, 1–6 layer-wise, which additionally trains only layers 1–6, shows superiority in several tasks, such as dependency parsing and important sentence extraction.

In addition, the models with two additional learnings, task-adaptive pre-training (TAPT) and domain-adaptive pre-training (DAPT), showed well-balanced performance in each task. TAPT performs additional learning on the textual data that is the target of the task, whereas DAPT performs additional learning in the domain to which the task belongs. The TAPT–DAPT combines the two.

In this study, we extracted the distributed representation of [CLS] tokens from three different models: 1–6 layer-wise, TAPT-only, and TAPT–DAPT, and concatenated the three vectors into a text feature. Here, CLS refers to classification embedding. It is often used as an embedding method for text classification with fine-tuning of the trained BERT models. In this study, the TAPT-only model is described as tapt512-60K, and the TAPT–DAPT model is described as dapt128-tapt512.

### 2.1.2. Emotion Corpus and Emotion Estimation Model

For emotion estimation, we used a model that was cross-trained using two types of corpora: the WRIME corpus [12] and our emotion corpus (Matsumoto corpus (https://github.com/Kmatsu-tokudai/emotionCorpusJapaneseTokushimaA2Lab) (accessed on 29 August 2022)) [13,14]. A breakdown of the two corpora is presented in Table 1. The definition of emotion class used in the WRIME corpus is based on Plutchik's basic emotions [15]: joy, sadness, anticipation, surprise, anger, fear, disgust, and trust. The Matsumoto corpus defined five emotion categories (joy, surprise, anger, sorrow, and neutral) based on Fisher's emotion chart [16].

**Table 1.** Breakdown of the emotional corpus.

| Corpus Type | Number of Emotion Class | Number of Sentences for Each Class | Total Number of Sentences |
|---|---|---|---|
| WRIME Corpus | 8-class (Plutchik's basic emotion) | Joy: 14,800 Sadness: 10,492 Anticipation: 6501 Surprise: 3885 Anger: 1648 Fear: 1910 Disgust: 1804 Trust: 1000 | 42,040 |
| Matsumoto Corpus | 5-class (based on Fischer's emotional chart) | Joy: 24,457 Surprise: 2903 Anger: 11,453 Sorrow: 20,622 Neutral: 16,966 | 76,401 |

The WRIME corpus is the corpus to which the respective emotions of the reader and writer is assigned. In this study, we integrated and used the emotions of both the readers and writers. Specifically, the label of the utterance was determined from the average of the two emotion vectors of the reader and writer.

Since the types of emotions assigned to the corpora are different, we built an emotion estimation model based on one corpus, estimated emotions using this emotion estimation model for the other corpus, and assigned the probability value of each output emotion class as an emotion feature vector. The training network for building the emotion estimation model is shown in Figure 1.

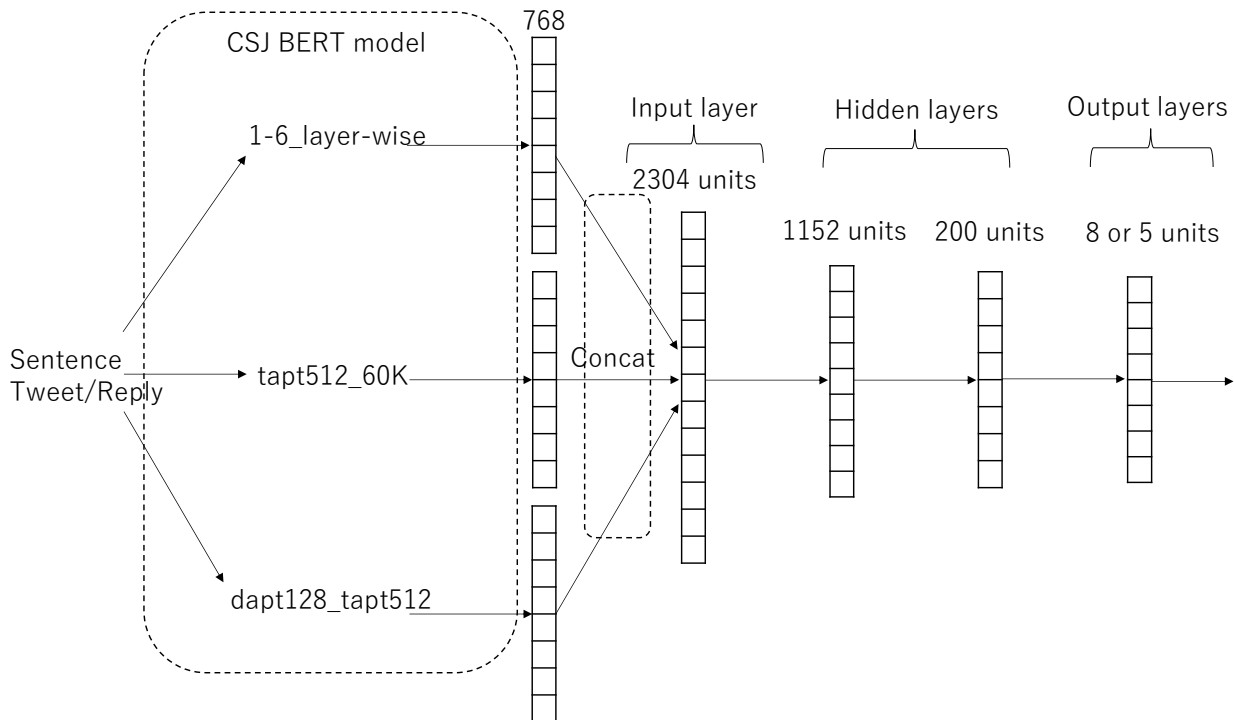

**Figure 1.** Emotion estimation networks for estimating eight emotions/five emotions.

Rectified linear unit (ReLU) was used for the middle layer activation function, and Softmax was used for the output layer activation function. The ReLU is an activation function with an output value of 0 when the input value is less than 0, and the output value is the same as the input value when the input value is over 0. Equation (1) shows the formula for the ReLU:

$$h(x) = \begin{cases} x(x > 0) \\ 0(x \le 0) \end{cases} \tag{1}$$

A dropout function was applied between the intermediate and middle layers with a dropout rate of 0.2. Categorical cross-entropy was used as the loss function during training, and Adam (adaptive moment) was used as the optimization method. Adam is a type of optimization algorithm for weight parameter values in neural networks and is commonly used in deep learning. Adaptive gradient (Adagrad) and root mean square propagation (RMSE), which are also commonly used, were improved to update the weights per parameter on a more appropriate scale by considering the mean square of the gradient and the mean as first- and second-order moments.

By doing this alternately for both corpora, we created a corpus with two types of emotion feature vectors, albeit pseudo-transparent. This combined corpus was trained using the multi-task learning model, shown in Figure 2. For each utterance in the corpus, we extracted and used 768 × 3 dimensional features using the CSJ BERT model, as described in Section 2.1.1. The activation function, dropout rate, and loss function were the same as those in the single-emotion estimation model, shown in Figure 1.

Due to the imbalance in the number of cases across classes in the training of the emotion estimation models in each corpus, we addressed the bias in the model outputs using oversampling methods, including the synthetic minority oversampling technique (SMOTE) [17] and Borderline SMOTE [18], as well as using undersampling methods, such as edited nearest neighbor (ENN) [19]. During training, the number of epochs was set to a maximum of 100, and the early stopping method was used to suppress overlearning. The ratio of the training data to the validation data was 8:2.

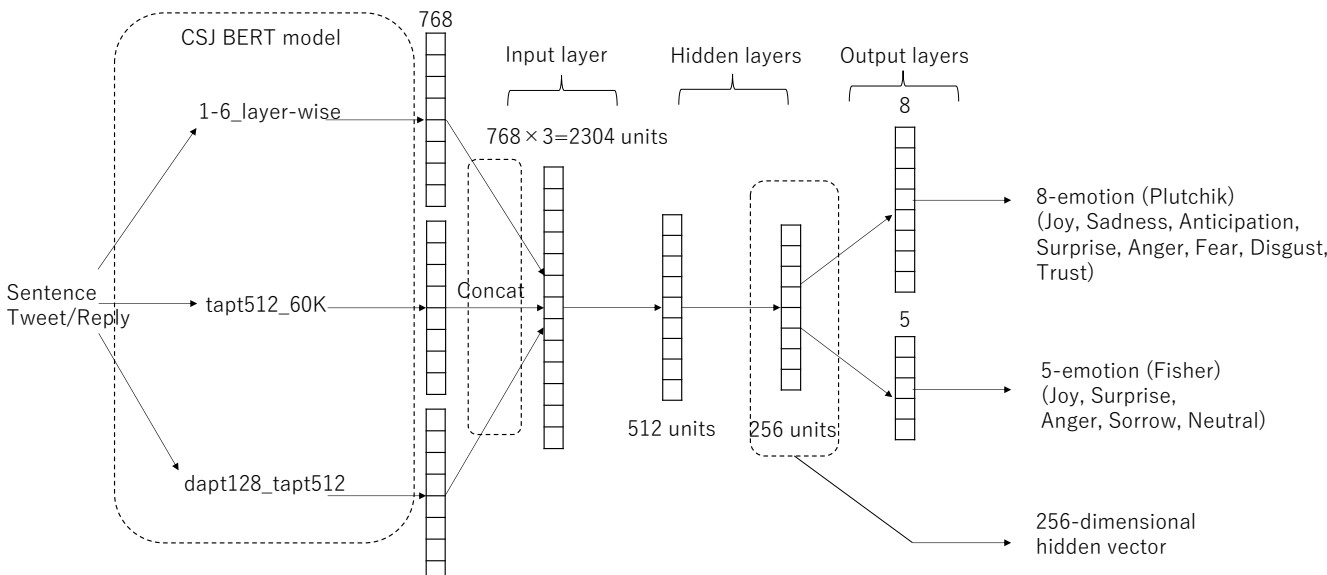

**Figure 2.** Emotion estimation network that outputs two types of emotion classes based on multi-task learning.

2.1.3. Image Feature Extraction from Pre-Trained Model

Image features were extracted from the pre-trained models for images attached to tweets. Below are the names of the networks used and the dimensionality of the features extracted from the intermediate layers.

Vgg16 [20]:                         512 dimensions
Resnet50 [21]:                     2048 dimensions
InceptionV3 [22]:             2048 dimensions
Xception [23]:                     2048 dimensions
DenseNet [24]:                    1024 dimensions
NASNet [25]:                       4032 dimensions
InceptionResNetV2 [26]:    1536 dimensions

The image size was converted to $128 \times 128$ pixels in batches and input into the pre-trained model to extract image features.

2.1.4. Conversion of Image to BERT Feature

To extract sensory features from images, a model that transforms images into BERT features was devised. Specifically, the relationship between image features and BERT features from caption text was learned using a neural network based on caption data assigned to images. To learn the model for converting images to BERT, we used "STAIR Captions" (https://www.wantedly.com/portfolio/projects/25771) (accessed on 29 August 2022) [27], and a large-scale image dataset "MS-COCO " to which caption texts in Japanese were assigned. This dataset consisted of 820,310 captions, each of which corresponded to one image in MS-COCO.

From the caption text in STAIR Captions, three types of 768-dimensional features were extracted using the CSJ BERT model introduced in Section 2.1.1. The neural network for training the transformation model had two intermediate layers (250 and 512 dimensions), and a hyperbolic tangent was used as the activation function between the intermediate layers. The activation function for the output layer was linear. The number of learning trials was set to 30 Epochs.

2.2. *Methodology for Predicting Tweet Reaction*

The proposed method predicts the number of "likes" and RTs obtained more than one week after a tweet was posted. The number of likes and RTs were varied and were

biased. Therefore, this method learned a model that predicts which quartile ranges they are included in by examining the overall distribution. The quartile range could be considered as a four-valued classification problem and consisted of four intervals: between the minimum and the first quartile, between the first and second quartiles, between the second and third quartiles, and between the third quartile and the maximum value.

The basic features were $768 \times 3$ dimensional features obtained from the tweet text (based on the CSJ BERT model), an average vector of $768 \times 3$ dimensional features obtained from the reply tweet text, and features obtained from the attached images (seven pre-trained models: Vgg16, Xception, Resnet50, InceptionV3, DenseNet, NASNet, and InceptionResNetV2). Furthermore, based on these features, the emotion estimation model was used to combine the two types of emotion vectors (8D, 5D) and the average vector of each feature vector (256D) obtained from the middle layer of the emotion-free model, and all were concatenated together.

## 3. Experiment and Results

In this section, we describe the evaluation experiments conducted to validate the proposed method and the results of the experiments. Section 3.1 describes the experimental conditions and data. and Section 3.2 describes the experimental results and compares them with those of the baseline method.

### 3.1. Experimental Setup

3.1.1. Layer Parameters and Training Parameters

Experiments were conducted using a five-part cross-validation method with stratified sampling. During oversampling, data far from the actual data might be produced. In this study, only 2247 cases were used in the dataset, and the distribution of the number of likes and RTs was skewed; thus, there were categories with an extremely small number of cases. Therefore, random oversampling was used. Undersampling was not used in this study because the total amount of training data became too small if the dataset was adjusted to the category with the fewest number of cases, and there was a risk of overtraining.

Optimizer: Adam
Loss function: Categorical cross-entropy
Epochs: 30
Resampling: Random undersampling

3.1.2. Dataset for Evaluation

This subsection describes the datasets used in the evaluation experiments.

The target was 2247 tweets, reply tweets, and attached images collected between December. 2020, and June, 2022. Twitter API version 2.0( https://developer.twitter.com/en/products/twitter-api) (accessed on 29 August 2022) [28] was used to collect the data. To collect historical data, we used an academic function (Academic Research product track (https://developer.twitter.com/en/products/twitter-api/academic-research) (accessed on 29 August 2022)) [29]. In addition, we set conditions under which the images had to be attached to the tweets and not be posted by bots. The number of likes and RTs were also obtained for the collected tweets. The distribution of the numbers of likes and RTs is shown in Figure 3.

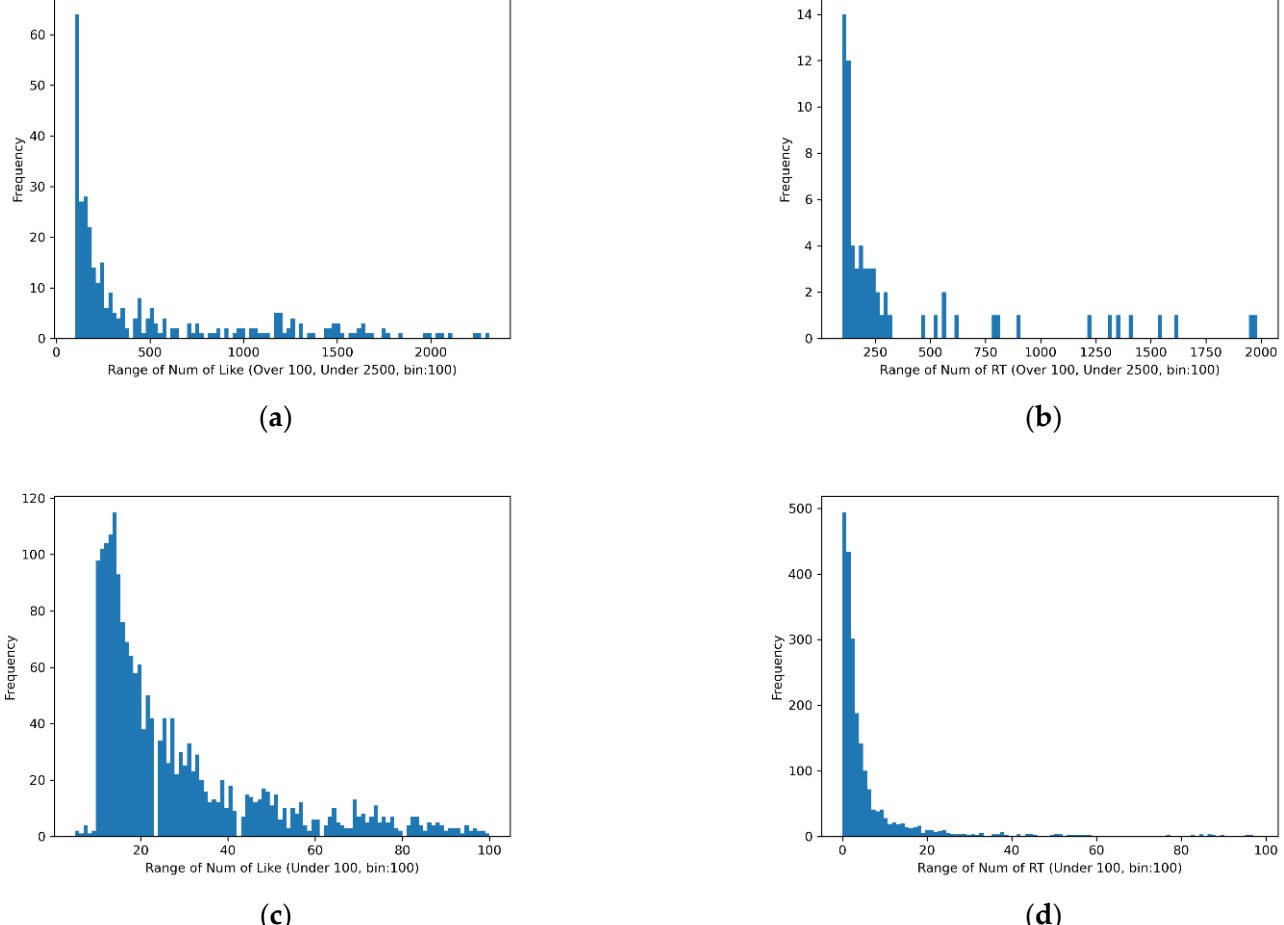

**Figure 3.** Distribution of number of likes and retweets; (**a**) number of likes (over100), (**b**) number of RTs (over100), (**c**) number of likes (under100), (**d**) number of RTs (under100).

### 3.1.3. Combinations of Feature

Table 2 shows the combination of features used in the proposed method. CSJ-BERT$_{tweet}$ and CSJ-BERT$_{reply}$ are three [CLS] token vectors concatenated from the tweet body or reply text using the Japanese spoken language BERT model, which has 2304 dimensions. IV$_x$ are image features extracted from the image using model x, which was pre-trained using the ImageNet database. Since the number of dimensions of this feature depends on the type of model x, the number of dimensions was denoted as D$_x$. EV$_w$, EV$_m$, and EV$_h$ are the emotion vectors of the two types of WRIME and Matsumoto corpora, where EV$_w$ represents the eight types of emotion vectors defined in the WRIME corpus, EV$_m$ represents the five types of emotion vectors defined in the Matsumoto corpus, and EV$_h$ represents the two types of emotions estimated in the WRIME and Matsumoto corpora. EV$_h$ is a 256-dimensional feature vector extracted from the hidden layer learned in the process of multi-task learning. The emotion estimation results differed for the three resampling methods applied: SMOTE, BorderlineSMOTE, and ENN. In this study, all the results were tested individually, and BordelineSMOTE was adopted as the one with the best performance.

**Table 2.** Combinations of feature used in the proposed methods.

| Method-ID | Without Affective Features | Dim. | With Affective Features | Dim. |
|---|---|---|---|---|
| A | CSJ-BERT$_{tweet}$ | 2304 | CSJ-BERT$_{tweet}$, EV$_w$, EV$_m$, EV$_h$ | 2573 |
| B | CSJ-BERT$_{reply}$ | 2304 | CSJ-BERT$_{reply}$, EV$_w$, EV$_m$, EV$_h$ | 2573 |
| C | IV$_x$ | D$_x$ | IV$_x$, EV$_w$, EV$_m$, EV$_h$ | D$_x$ + 269 |
| AB | CSJ-BERT$_{tweet}$, CSJ-BERT$_{reply}$ | 4608 | CSJ-BERT$_{tweet}$, CSJ-BERT$_{reply}$ EV$_w$, EV$_m$, EV$_h$ | 5415 |
| CA | IV$_x$, CSJ-BERT$_{tweet}$ | D$_x$ + 2304 | IV$_x$, CSJ-BERT$_{tweet}$, EV$_w$, EV$_m$, EV$_h$ | D$_x$ + 2573 |
| CB | IV$_x$, CSJ-BERT$_{reply}$ | D$_x$ + 2304 | IV$_x$, CSJ-BERT$_{reply}$, EV$_w$, EV$_m$, EV$_h$ | D$_x$ + 2573 |
| ABC | CSJ-BERT$_{tweet}$, CSJ-BERT$_{reply}$, IV$_x$ | 4608 + D$_x$ | CSJ-BERT$_{tweet}$, CSJ-BERT$_{reply}$, IV$_x$, EV$_w$, EV$_m$, EV$_h$ | 5415 + D$_x$ |

### 3.1.4. Baseline Methods

Two machine learning algorithms, the support vector machine and random forest classifier, were used as baseline methods with all features except the affective feature. To adjust for the bias in the amount of data per class, we used random oversampling, as in the proposed method. The support vector machine and random forest classifier utilized the classes included in scikit-learn (https://scikit-learn.org/stable/) (accessed on 29 August 2022) [30]. A pre-trained word distributed representation (hottoSNS-w2v (https://github.com/hottolink/hottoSNS-w2v) (accessed on 29 August 2022 [31,32]) was used to extract features from the text of tweets and replies to tweets using the baseline method. This is a large-scale distributed representation model trained by HottoLink Inc., based on Japanese Wikipedia and Japanese social networking services (SNS) text data. The vocabulary was approximately two million words. There are two types of distributed representation models: a 100-dimensional model based on Wikipedia and a 200-dimensional model based on SNS.

The baseline method calculated the average vector of the distributed expression vectors of words in a sentence and used it as the feature vector of the text. The same features as those in the proposed method were used for image features. We also used SimpleTransformers (https://simpletransformers.ai/) (accessed on 29 August 2022) [33] and two different methods as a baseline: tweets only and tweets and image embedding as inputs. For the pre-trained embedding model, we used the BERT model published by Tohoku University (https://github.com/cl-tohoku/bert-japanese) (accessed on 29 August 2022) [34].

Table 3 shows the method IDs, corresponding features, and number of dimensions used in the baseline method. In this table, Wiki$_{tweet}$ and Wiki$_{reply}$ were the average vectors of 100-dimensional word embedding vectors obtained from words in the tweet and reply texts, respectively, using Word2vec's word-distributed representation model, based on Japanese Wikipedia.

**Table 3.** Combinations of features used in the baseline methods.

| Method-ID | Combination of Features | Dim. |
|---|---|---|
| $\alpha_w$ | Wiki$_{tweet}$ | 100 |
| $\alpha_s$ | SNS$_{tweet}$ | 200 |
| $\beta_w$ | Wiki$_{reply}$ | 100 |
| $\beta_s$ | SNS$_{reply}$ | 200 |
| $\gamma$ | IV$_x$ | D$_x$ |
| $\alpha_w \beta_w$ | Wiki$_{tweet}$, Wiki$_{reply}$ | 200 |
| $\alpha_s \beta_s$ | SNS$_{tweet}$, SNS$_{reply}$ | 400 |
| $\gamma \alpha_w$ | IV$_x$, Wiki$_{tweet}$ | D$_x$ + 100 |
| $\gamma \alpha_s$ | IV$_x$, SNS$_{tweet}$ | D$_x$ + 200 |
| $\gamma \beta_w$ | IV$_x$, Wiki$_{reply}$ | D$_x$ + 100 |
| $\gamma \beta_s$ | IV$_x$, SNS$_{reply}$ | D$_x$ + 200 |
| $\alpha_w \beta_w \gamma$ | Wiki$_{tweet}$, Wiki$_{reply}$, IV$_x$ | 300 + D$_x$ |
| $\alpha_s \beta_s \gamma$ | SNS$_{tweet}$, SNS$_{reply}$, IV$_x$ | 300 + D$_x$ |
| Transformer$_{tweet}$ | BERT$_{tweet}$ | 768 × 512 |
| Multi-modal Transformer | BERT$_{tweet}$ + Image Embedding | 768 × 512 + D$_x$ |

All the evaluation experiments used a five-fold cross-validation method. To ensure fairness, the splitting of the training and test data was the same for all experiments.

### 3.2. Experimental Result

Tables 4–7 show the accuracy, weighted averaged accuracy, macro-averaged accuracy, and F1 score for each interquartile range of likes/RTs. Since there are many feature and algorithm combinations, this study presents only the top 10 results in terms of accuracy. The 'Acc.' column indicates accuracy, the 'macro' column indicates macro averaged accuracy, and the 'weighted' column indicates weighted averaged accuracy.

**Table 4.** Top 10 F1-scores and accuracies of like prediction (proposed methods).

| Method-ID | Feature | Emotion | F1-Scores for Each Quartile Range of Likes | | | | Acc. | Macro | Weighted |
|---|---|---|---|---|---|---|---|---|---|
| | | | 0–0.25 | 0.25–0.5 | 0.5–0.75 | 0.75–1.0 | | | |
| B | $BERT_{reply}$ | Y | 0 | 0.870 | 0.092 | 0.205 | 0.602 | 0.774 | 0.442 |
| B | $BERT_{reply}$ | N | 0 | 0.871 | 0.094 | 0.253 | 0.579 | 0.774 | 0.449 |
| CA | $BERT_{tweet} + IV_{vgg16}$ | Y | 0 | 0.916 | 0.122 | 0.197 | 0.564 | 0.843 | 0.450 |
| CA | $BERT_{tweet} + IV_{vgg16}$ | N | 0 | 0.916 | 0.113 | 0.197 | 0.561 | 0.843 | 0.447 |
| CA | $BERT_{tweet} + IV_{densenet121}$ | N | 0 | 0.918 | 0.098 | 0.199 | 0.524 | 0.845 | 0.435 |
| CA | $BERT_{tweet} + IV_{xception}$ | Y | 0 | 0.919 | 0.112 | 0.156 | 0.508 | 0.848 | 0.424 |
| CA | $BERT_{tweet} + IV_{xception}$ | N | 0 | 0.918 | 0.111 | 0.126 | 0.506 | 0.847 | 0.415 |
| CA | $BERT_{tweet} + IV_{densenet121}$ | Y | 0 | 0.917 | 0.104 | 0.203 | 0.505 | 0.843 | 0.432 |
| CA | $BERT_{tweet} + IV_{inceptionresnetv2}$ | N | 0 | 0.921 | 0.094 | 0.188 | 0.471 | 0.852 | 0.419 |
| CA | $BERT_{tweet} + IV_{resnet50}$ | N | 0 | 0.921 | 0.084 | 0.168 | 0.466 | 0.852 | 0.410 |

**Table 5.** Top 10 F1-scores and accuracies of RT prediction (proposed methods).

| Method-ID | Feature | Emotion | F1-Scores for Each Quartile Range of RTs | | | | Acc. | Macro | Weighted |
|---|---|---|---|---|---|---|---|---|---|
| | | | 0–0.25 | 0.25–0.5 | 0.5–0.75 | 0.75–1.0 | | | |
| AB | $BERT_{tweet} + BERT_{reply}$ | N | 0.000 | 0.933 | 0.000 | 0.000 | 0.996 | 0.991 | 0.332 |
| C | $IV_{vgg16}$ | N | 0.000 | 0.928 | 0.000 | 0.000 | 0.996 | 0.991 | 0.332 |
| AB | $BERT_{tweet} + BERT_{reply}$ | Y | 0.000 | 0.931 | 0.000 | 0.000 | 0.995 | 0.991 | 0.332 |
| C | $IV_{densenet121}$ | Y | 0.000 | 0.000 | 0.000 | 0.000 | 0.995 | 0.991 | 0.332 |
| A | $BERT_{tweet}$ | N | 0.000 | 0.930 | 0.000 | 0.000 | 0.995 | 0.990 | 0.332 |
| A | $BERT_{tweet}$ | Y | 0.000 | 0.930 | 0.000 | 0.000 | 0.995 | 0.990 | 0.332 |
| C | $IV_{densenet121}$ | N | 0.000 | 0.928 | 0.000 | 0.000 | 0.995 | 0.990 | 0.332 |
| C | $IV_{xception}$ | N | 0.000 | 0.928 | 0.000 | 0.000 | 0.995 | 0.990 | 0.332 |
| C | $IV_{xception}$ | Y | 0.000 | 0.000 | 0.000 | 0.000 | 0.995 | 0.990 | 0.332 |
| C | $IV_{vgg16}$ | Y | 0.000 | 0.000 | 0.000 | 0.000 | 0.995 | 0.990 | 0.332 |

**Table 6.** Top 10 F1-scores and accuracies of like prediction (baseline methods).

| Method-ID | Feature | Emotion | F1-Scores for Each Quartile Range Of Likes | | | | Acc. | Macro | Weighted |
|---|---|---|---|---|---|---|---|---|---|
| | | | 0–0.25 | 0.25–0.5 | 0.5–0.75 | 0.75–1.0 | | | |
| $SNS_{reply}$ (RF) | $\beta_s$ | N | 0.460 | 0.030 | 0.200 | 0.570 | 0.380 | 0.320 | 0.310 |
| $Wiki_{reply}$ (RF) | $\beta_w$ | N | 0.450 | 0.020 | 0.210 | 0.570 | 0.380 | 0.310 | 0.310 |
| $Wiki_{reply}$ (SVM) | $\beta_w$ | N | 0.460 | 0.000 | 0.000 | 0.570 | 0.380 | 0.260 | 0.250 |
| $Wiki_{tweet} + Wiki_{reply}$ (RF) | $\alpha_w \beta_w$ | N | 0.368 | 0.316 | 0.229 | 0.551 | 0.380 | 0.366 | 0.366 |
| $Wiki_{tweet} + Wiki_{reply}$ (SVM) | $\alpha_w \beta_w$ | N | 0.455 | 0.000 | 0.000 | 0.561 | 0.373 | 0.254 | 0.250 |
| $SNS_{reply}$ (SVM) | $\beta_s$ | N | 0.460 | 0.000 | 0.060 | 0.570 | 0.370 | 0.270 | 0.270 |
| $SNS_{tweet}$ (SVM) | $\alpha_s$ | N | 0.290 | 0.250 | 0.320 | 0.310 | 0.290 | 0.290 | 0.290 |
| $Wiki_{tweet} + IV_{nasnet121}$ (SVM) | $\alpha_w \beta_w \gamma$ | N | 0.154 | 0.167 | 0.389 | 0.305 | 0.287 | 0.254 | 0.273 |
| $SNS_{tweet}$ (RF) | $\alpha_s$ | N | 0.300 | 0.260 | 0.260 | 0.320 | 0.280 | 0.280 | 0.280 |
| $Wiki_{tweet} + IV_{xception}$ (SVM) | $\alpha_w \beta_w \gamma$ | N | 0.008 | 0.414 | 0.015 | 0.143 | 0.276 | 0.145 | 0.149 |

**Table 7.** Top 10 F1-scores and accuracies of RT prediction (baseline methods).

| Method-ID | Feature | Emotion | F1-Scores for Each Quartile Range of RTs | | | | Acc. | Macro | Weighted |
|---|---|---|---|---|---|---|---|---|---|
| | | | 0–0.25 | 0.25–0.5 | 0.5–0.75 | 0.75–1.0 | | | |
| Wiki$_{tweet}$ + Wiki$_{reply}$ (RF) | $\alpha_w \beta_w$ | N | 0.229 | 0.129 | 0.363 | 0.437 | 0.321 | 0.289 | 0.308 |
| SNS$_{reply}$ (RF) | $\beta_s$ | N | 0.380 | 0.050 | 0.170 | 0.450 | 0.320 | 0.260 | 0.270 |
| Wiki$_{reply}$ (SVM) | $\beta_w$ | N | 0.380 | 0.000 | 0.000 | 0.470 | 0.320 | 0.210 | 0.210 |
| Wiki$_{tweet}$ + Wiki$_{reply}$ (SVM) | $\alpha_w \beta_w$ | N | 0.393 | 0.009 | 0.000 | 0.470 | 0.316 | 0.218 | 0.211 |
| SNS$_{tweet}$ (RF) | $\alpha_s$ | N | 0.220 | 0.130 | 0.420 | 0.310 | 0.310 | 0.270 | 0.290 |
| Wiki$_{reply}$ (RF) | $\beta_w$ | N | 0.390 | 0.030 | 0.160 | 0.420 | 0.310 | 0.250 | 0.260 |
| SNS$_{reply}$ (SVM) | $\beta_s$ | N | 0.380 | 0.040 | 0.050 | 0.470 | 0.310 | 0.230 | 0.230 |
| Wiki$_{tweet}$ (RF) | $\alpha_w$ | N | 0.210 | 0.160 | 0.390 | 0.270 | 0.290 | 0.260 | 0.270 |
| Wiki$_{tweet}$ (SVM) | $\alpha_w$ | N | 0.250 | 0.140 | 0.410 | 0.170 | 0.290 | 0.240 | 0.260 |
| SNS$_{tweet}$ (SVM) | $\alpha_s$ | N | 0.250 | 0.210 | 0.340 | 0.300 | 0.280 | 0.280 | 0.290 |

The calculation formulae for accuracy, weighted averaged accuracy, macro-averaged accuracy recall, precision, and F1-score are shown in Equations (2)-(5). In Equation (2), $y_i$ represents the correct class of data, $\hat{y}_i$ represents the estimated class, $N$ represents the number of data to be covered.

$L$ in Equations (3) and (4) represents the set of class, and $|L|$ represents the number of classes. The set of class is $L = \{'0 - 0.25', '0.25 - 0.5', '0.5 - 0.75', '0.75 - 1.0'\}$. Each label means quartile range of likes/RTs.

In Equation (5), $y_l$ represents the set of data in which the correct class is $l$, and $\hat{y}_l$ represents the set of data in which the estimated class is $l$.

$$\text{Accuracy}(y, \hat{y}) = \frac{1}{N-1} \sum_{i=0}^{N-1} (\hat{y}_i = y_i) \tag{2}$$

$$\text{Weighted Averaged Accuracy} = \frac{1}{\sum_{l \in L} |y_l|} \sum_{l \in L} |y_l| \times \text{Accuracy}(y_l, \hat{y}_l) \tag{3}$$

$$\text{Macro Averaged Accuracy} = \frac{1}{|L|} \sum_{l \in L} \text{Accuracy}(y_l, \hat{y}_l) \tag{4}$$

$$\begin{aligned} \text{Recall}_l &= \frac{|y_l \cap \hat{y}_l|}{|y_l|} \\ \text{Precision}_l &= \frac{|y_l \cap \hat{y}_l|}{|\hat{y}_l|} \\ \text{F1-Score}_l &= \frac{2 \times \text{Recall}_l \times \text{Precision}_l}{\text{Recall}_l + \text{Precision}_l} \end{aligned} \tag{5}$$

A, B, C, CA, CB, and ABC are method IDs listed in Table 2. The letters 'Y and 'N' in the 'Emotion' column indicate whether the affective feature was used (Y) or not (N).

The results show that the reply feature yielded a higher percentage of correct responses than other features. In Table 4, the accuracy was higher when using the affective feature, indicating that the affective feature was effective to some extent. Furthermore, the estimation performance of the image features alone was not high; however, when the image feature Vgg16 was combined with the tweet feature, it showed high performance. Table 5 shows that all the accuracies were almost the same, and the F1-score for the RT quartile range of 0.25–0.5 tended to be almost zero. However, the F1-score was generally low, indicating that RT was more difficult to estimate than the like. The distribution trend of the number of RTs suggested that this was partly due to the large bias in the number of cases in each RT quartile.

The baseline method using image features was not in the top 10 for either likes or RTs, and although the accuracy was low, it obtained an F1-score above 0.4 in the range of 0.75–1.0, and some F1-score in the range of 0.5–0.75. The F1-score was obtained in the range of 0.5–0.75.

These baseline methods used classification models trained by random forest and SVM on the mean vector of word variance representations using word2vec, which was trained based on Wikipedia. It could be observed that trend detection was possible even

without using a complex model, such as the transformer, if the meaning of words could be estimated, even partially.

### 3.3. Additional Experiment

Here, we analyzed the relationship between the emotional similarity of tweets and replies and the number of likes/RTs by visualizing them. We transformed tweets and replies to tweets into four vectors: a CSJ-BERT vector, a vector of eight emotion estimation results, a vector of five emotion estimation results, and a feature vector extracted from the hidden layer of the emotion estimation model. We then calculated the degree of similarity between them. The obtained similarities were plotted on the two-dimensional coordinate axes, taking each of them as an axis. In addition, the like-similarity and RT-similarity relationships were visualized. Figure 4a–h shows the results. The results show that there was no clear relationship between like and RT among these similarities:

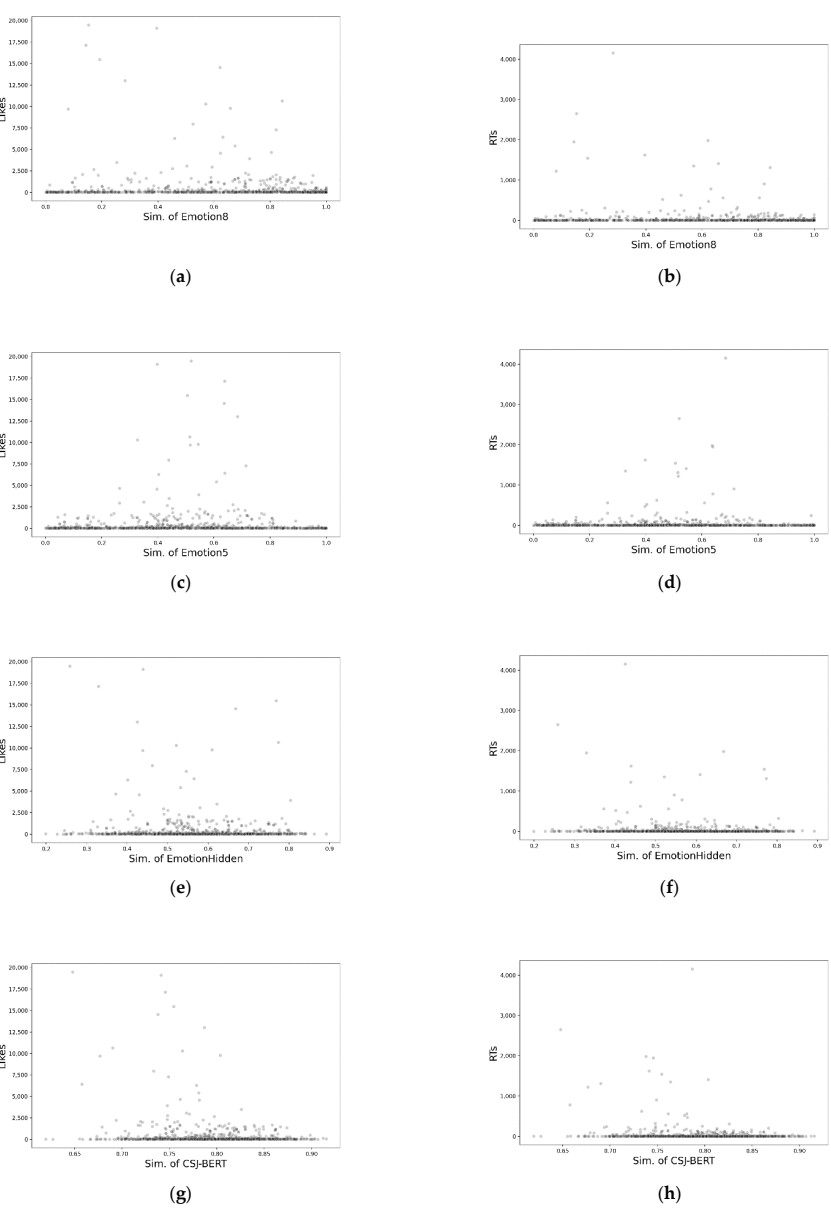

**Figure 4.** Relation between like/RT and feature vectors of tweets and replies. (**a**) Relation of $sim_{tw8,rp8}$ and like, (**b**) Relation of $sim_{tw8,rp8}$, RT, (**c**) Relation of $sim_{tw5,rp5}$, like, (**d**) Relation of $sim_{tw5,rp5}$, RT, (**e**) Relation of $sim_{twh,rph}$, like, (**f**) Relation of $sim_{twh,rph}$, RT, like, (**g**) Relation of $sim_{twb,rpb}$, like, (**h**) Relation of $sim_{twb,rpb}$, RT.

$\text{sim}_{\text{tw8,rp8}}$ ... Similarity between the eight emotion vectors of tweets and replies
$\text{sim}_{\text{tw5,rp5}}$ ... Similarity between the five emotion vectors of tweets and replies
$\text{sim}_{\text{twh,rph}}$ ... Similarity between the hidden emotion vectors of a tweet and reply
$\text{sim}_{\text{twb,rpb}}$ ... Similarity between CSJ-BERT vectors of tweets and replies

The distribution of emotional similarity in the RT and like quartile ranges is shown in Figure 5. The brighter the color, the more frequently the data were estimated as a combination of their like and RT labels. We added an explanation for this. The labels 0-0.25, 0.25-0.5, 0.5-0.75, 0.75-1.0 indicate the position in the quartile range, and a higher number indicated a higher number of likes and RTs. In this figure, the mean value of similarity was obtained for each RT, similar to the quartile range combination, and shown as a heat map. From this figure, it can be observed that emotional similarity tended to be higher when RTs and likes were high. This result confirmed that information diffusion occurs when a user posts a reply to a tweet with empathy, which is then retweeted or liked by another user who has read the reply.

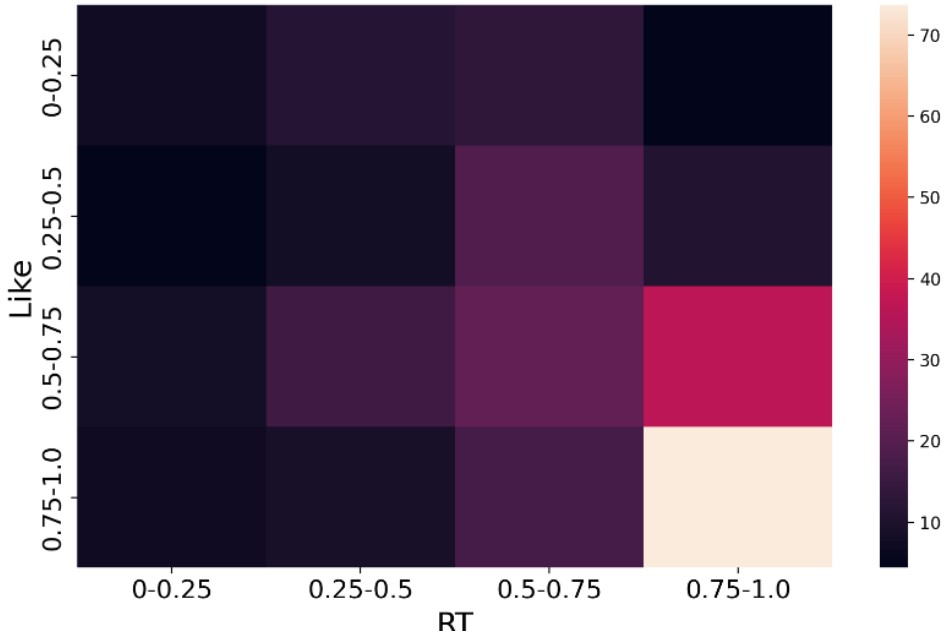

**Figure 5.** Relationship between RT, like, and emotion vectors (8 emotions).

Next, the majority vote of the results of the affective estimation for tweets and replies was calculated for each tweet. A violin plot for each quartile range of the number of likes and RTs is shown in Figure 6. For the eight emotions based on the WRIME corpus, both likes and RTs were biased toward joy as the values increased. For the five emotions based on the Matsumoto corpus, there was a wide distribution toward joy and neutral, regardless of the values of likes and RTs. Characteristically, the larger the value of both likes and RTs, the more tweets and replies were categorized as neutral. This might indicate that it is difficult to detect neutral sentiment in many tweets and replies, rather than that the estimation of the eight emotions was not successful.

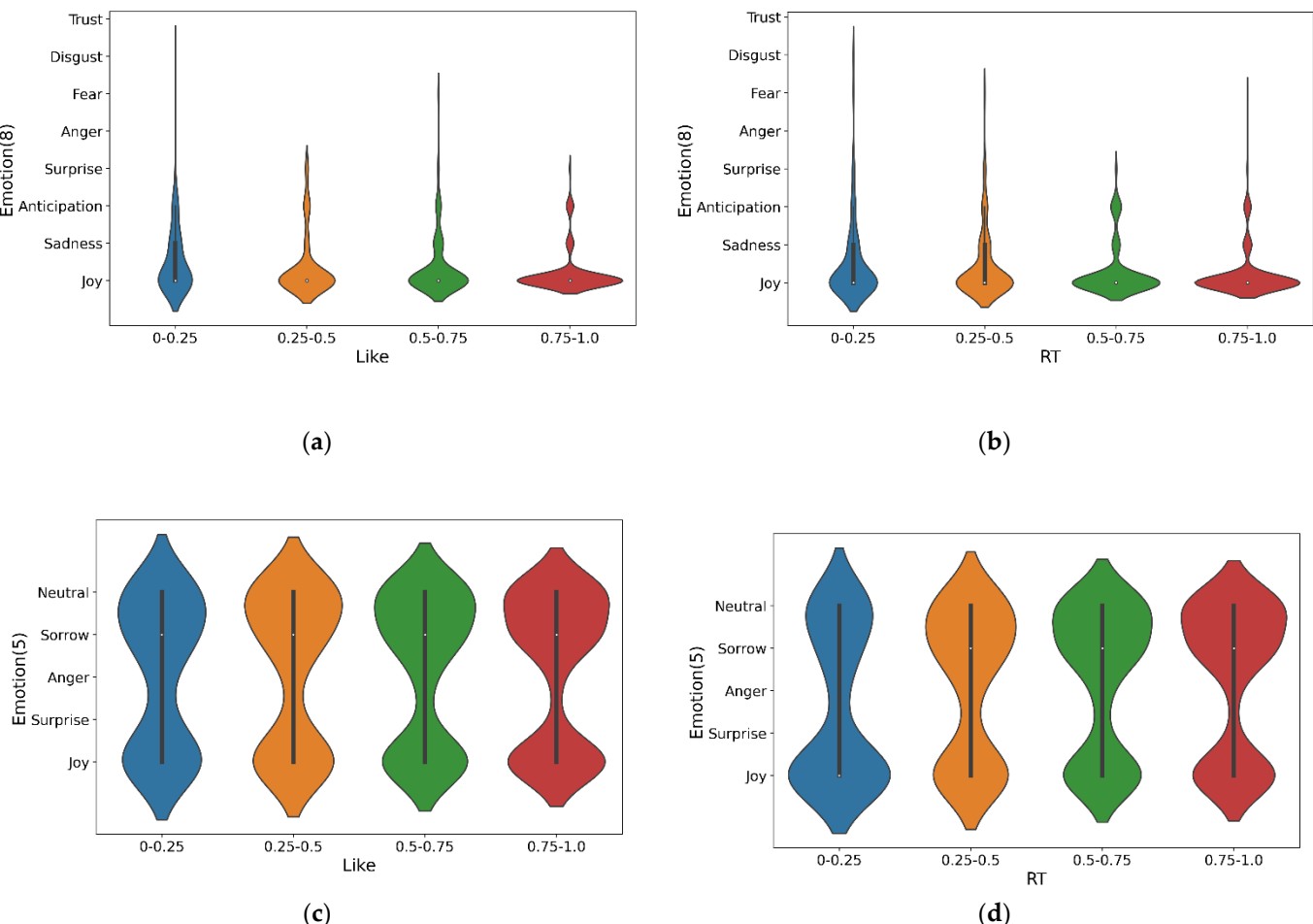

**Figure 6.** Relationship between RT, like, and tweet and reply emotions; (**a**) eight emotions and likes, (**b**) eight emotions and RTs, (**c**) five emotions and likes, (**d**) five emotions and RTs (the horizontal label indicates the quartile range label of like/RT).

## 4. Discussion

Among the proposed methods, replies were effective features when evaluated based on accuracy. However, replies could not be used to predict the buzz phenomenon because they were assigned after a tweet was posted and did not contain information that could be obtained before the tweet was posted. We found that the proposed method, which combines image features and tweet features, showed good performance for the prediction of likes. The affective estimation results did not have a significant impact on performance. However, we believe that the affective features obtained from tweets, replies, and images provided clues for classifying tweets.

Another factor contributing to the improved performance of the combination of image features and tweet features is that, in the case of tweets with images, it is often difficult to convey the meaning using only linguistic information, and the meaning can be supplemented by image information. This is also true for image-only tweets, but in general, image-only tweets do not generate much buzz. This is because it is often difficult to grasp the meaning of an image alone. In some cases, when text is included in an image, meaning may be sufficiently conveyed. However, processing images to add text is not often used because it would spoil the convenience and real-time nature unique to Twitter, such as taking a picture and tweeting it. Therefore, a combination of text and image features was considered a promising feature set.

In addition, by considering the relationship between the emotions of tweets and replies, we found that both RTs and likes were higher when similar emotions were expressed.

This can be interpreted as the result of information diffusion caused by the interaction between empathy and SNS. To predict this in advance, it is necessary to predict replies from information in posted tweets and attached images. If replies can be predicted, their emotions can be estimated; thus, the similarity of affective features can also be obtained. However, predicting replies from posted tweets is difficult. Some response generation models used in dialogue systems collect a large number of SNS exchanges (the relationship between tweets and their replies) as examples of dialogue responses and use them as training data. By incorporating these methodologies, it is possible to predict replies, from the content of tweets, and trends, based on the information in the replies.

## 5. Conclusions

In this study, we proposed a model that predicts the number of "likes" and "RTs" using the content of tweets and attached images as features to predict the buzz phenomenon on Twitter when tweets are posted. Since the buzz phenomenon often occurs when many users empathize with the content of a post, we investigated a method to improve prediction accuracy by adding emotions that can be estimated from the tweet content and images as features. As an algorithm for emotion estimation, we used a model in which a text corpus with emotion labels was converted into a feature vector using a pre-training model of BERT for spoken Japanese and trained by a deep neural network. The model was trained using a corpus constructed by the authors and another corpus with emotion labels constructed in a previous study, aiming at emotion estimation from various perspectives. As it is difficult to estimate emotions directly from images, a neural network was trained to estimate the features of spoken Japanese BERT from image features using a dataset of captions for images, and, thus, indirectly achieving emotion estimation from images.

We demonstrated the effectiveness of the proposed method by comparing it with a baseline method that classifies word embedding and image features using SVM and random forest machine-learning algorithms. In addition, the use of reply text, which is the information after a tweet is posted, sometimes results in a high percentage of correct responses. Elements such as positive feedback, in replies may have a strong influence on the diffusion of information, so we believe that a method to quickly detect the diffusion of information by analyzing the emotions of replies posted at an early stage would be useful.

However, when emotional features are used to predict trends, it is necessary to determine whether the perspective is that of the reader or writer. In the future, we would like to study a method for predicting trends using emotional features, including an analysis of the subjectivity of the tweet content.

**Author Contributions:** Conceptualization, K.M.; Data curation, R.A.; Funding acquisition, K.M. and M.Y.; Methodology, K.M. and R.A.; Supervision, K.K.; Validation, M.Y.; Visualization, K.M.; Writing—original draft, K.M.; Writing—review & editing, K.M., M.Y. and K.K. All authors have read and agreed to the published version of the manuscript.

**Funding:** This work was supported by the 2022 SCAT Research Grant and JSPS KAKENHI Grant Number JP20K12027, JP21K12141.

**Institutional Review Board Statement:** Not applicable.

**Informed Consent Statement:** Not applicable.

**Data Availability Statement:** Not applicable.

**Conflicts of Interest:** The authors declare no conflict of interest.

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
