# Peer review of "Trend Prediction Based on Multi-Modal Affective Analysis from Social Networking Posts"

_electronics, doi:10.3390/electronics11213431_

Round 1

Reviewer 1 Report

In my opinion, the work is well written. It concerns current phenomena. Information noise is ubiquitous, so all methods of influencing what we want to display are useful.

Reviewer 2 Report

This paper proposed a multi-modal method to forecasting the affection of posts during their early stage. In general, the paper is well-structured and well-written, the proposed method is new to the field and is an interesting attempt. There are only a few minor issues need to improve before its publication.

P9, 296, the metrics, such as accuracy, F1-score, etc., should be explained in more detail.

Figure 4 and 5, what does “Maximum score of each proposed method for like prediction” mean, why only report the maximum score? In fact, it is not straightforward to read these figures, from fig. 4 to 8, I recommend to integrate the results into a few tables.

P13, figure 9, only the first subfigure has the number, and these figures don’t really mean much.

P14, 379, I don’t really find significant improvements in fact, do authors can tune down the claims.

Reviewer 3 Report

The manuscript entitled: "Trend Prediction based on Multi-modal Affective Analysis from Social Networking Posts" is relevant for the Electronics journal. The article is based on original experimental research. Overall, the paper is well prepared. Nevertheless, the article required minor improvements (in the attachment).
